# Utility of carotid ultrasound on prediction of 1-year mortality in emergency department patients with neurological deficits: A 10-year population-based cohort study

**Chi-Hsin Chen[1], Chih-Wei Sung[1], Jiann-Shing Jeng[2], Cheng-Yi Fan[1], Jia-How Chang[1], Jiun-Wei Chen[1], Sung-Chun Tang[2]ʘ\*, Edward Pei-Chuan Huang**(ID)[1,3]ʘ\*

**1** Department of Emergency Medicine, National Taiwan University Hospital Hsin-Chu Branch, Hsinchu City, Taiwan, **2** Stroke Center & Department of Neurology, National Taiwan University Hospital, Taipei City, Taiwan, **3** Department of Emergency Medicine, National Taiwan University Hospital, Taipei City, Taiwan

ʘ These authors contributed equally to this work.

\* tangneuro@gmail.com (SCT); edward56026@gmail.com (EPCH)

**Data Availability Statement:** Data cannot be shared publicly because of ethics issues. Data are available from Ethics Committee of National Taiwan

## Abstract

### Background

This study aimed to investigate the association between the carotid ultrasound results and 1-yr mortality of patients with neurological deficits in the emergency department (ED).

### Methods

This study included patients with neurological symptoms who presented to the ED between January 1, 2009 and December 31, 2018, and underwent sonographic imaging of the bilateral carotid bulb, common carotid artery (CCA), internal carotid artery (ICA), and external carotid arteries. A stenosis degree of >50% was defined as significant carotid stenosis. Carotid plaque score (CPS) was calculated by adding the score of stenosis severity of all segments. The association between carotid ultrasound results and 1-yr mortality was investigated using the Cox regression model.

### Results

The analysis included 7,961 patients (median age: 69 yr; men: 58.7%). Among them, 247 (3.1%) passed away from cardiovascular (CV)-related causes, and 746 (9.4%) died within a year. The mortality group presented with more significant carotid stenosis of the carotid bulb, CCA, or ICA and had a higher median CPS. A higher CPS was associated with a greater 1-yr all-cause mortality (adjusted hazard ratio [aHR] = 1.08; 95% confidence interval [CI] = 1.03–1.13; $p$ = 0.001; log-rank $p$ < 0.001) and CV-related mortality (aHR = 1.13; 95% CI = 1.04–1.22; $p$ = 0.002, log-rank $p$ < 0.001). Significant stenosis of either carotid artery segment did not result in a higher risk of 1-yr mortality.

University Hospital (contact via https://www.ntuh.gov.tw/RECO/Index.action) for researchers who meet the criteria for access to confidential data.

**Funding:** Grant recipient: Edward Pei-Chuan Huang Grant number: 110-S5085 Funder: National Taiwan University Hospital URL: https://www.ntuh.gov.tw/ntuh/Index.action?l=en_US ==== Funder: National Taiwan University Hospital Hsin-Chu Branch (111-HCH049) Grant recipient: Chi-Hsin Chen. The funders had no role in study design, data collection and analysis, decision to publish, or preparation of the manuscript.

**Competing interests:** The authors have declared that no competing interests exist.

## Conclusions

We comprehensively investigated the utility of carotid ultrasound parameters on predicting mortality in this 10-yr population-based cohort, which included over 7,000 patients with acute neurological deficits presented to the ED. The result showed that CPS could be used as risk stratification tools for 1-yr all-cause and CV mortality.

## Introduction

Acute neurological deficit is a common chief complaint that leads patient to emergency department (ED) visits [1], is mostly caused by ischemic events like ischemic stroke or transient ischemic attack (TIA), and has been the leading cause of mortality [2, 3]. Chen et al. reported 3% and 16% 28-day and 5-yr mortality rates after ischemic stroke, respectively [4]. Cardiovascular (CV) events, such as cerebrovascular accidents and ischemic heart disease, are the main causes of mortality in such patients. These conditions have a similar risk factor and pathogenesis, which is atherosclerosis [5–7].

Carotid artery atherosclerosis accounts for approximately 15%–20% of ischemic stroke cases [8]. Carotid ultrasonography, one of several imaging modalities, is a noninvasive, contrast-free, and radiation-free method for screening carotid and vertebral artery atherosclerosis. As a result, it is currently widely used for etiological surveys and risk stratification among patients with focal neurological deficits, especially those who were suspected of ischemic stroke [9, 10]. Previous studies have shown an association between carotid stenosis of ≥50% and a higher risk of recurrent stroke [11]. Carotid plaque score (CPS), which is obtained by adding the total plaque severity scores of each carotid segment, can also reflect the plaque burden and atherosclerotic severity in addition to carotid artery segment stenosis [12]. Major CV diseases were more likely to develop in patients with higher CPS [12–18]. According to the Rotterdam study, the risk of different stroke subtypes may increase with a higher plaque score [13]. In addition, CPS is considered as a strong coronary artery disease (CAD) indicator [16, 17]. A meta-analysis revealed CPS as likely the best carotid ultrasound CAD predictor, and it has a higher diagnostic accuracy than intima-media thickness, which is previously widely used [18].

Carotid ultrasound is recommended as a primary examination in patients with suspected ischemic strokes who visit the ED as it enables timely interventions and guides secondary prevention in patients with significant carotid atherosclerosis [19]. Studies that investigate the association between carotid ultrasound results and long-term mortality and cardiovascular mortality, however, were relatively rare. Ogren et al. reported an excess higher 10-yr CV mortality in 470 men aged 68 yr with asymptomatic carotid stenosis [20]. According to the Tromsø study, individuals with carotid stenosis have a mortality risk that is more than three times higher [21]. Some previous studies reported that a higher total plaque burden or CPS increased the risk of mortality in elderly patients or those with hypertension [22–24]. Neither of the above studies focused on patients in the ED with neurological deficits, whose risk of carotid stenosis and fatal CV events may be even higher.

Therefore, this study aimed to explore the association between the carotid ultrasonography results, including carotid stenosis and CPS, and 1-yr mortality among patients with neurological symptoms who visit the ED. The objective is to investigate the utility and the optimal parameter of carotid ultrasound on prediction of mortality. The results of the current study may be an important guide for risk stratifications in ED practice.

## Materials and methods

### Study design and setting

A 10-yr retrospective cohort study was conducted on patients with neurological symptoms who visited the ED of the National Taiwan University Hospital (NTUH) between January 1, 2009 and December 31, 2018. Patient data, including demographic characteristics, previous comorbidities, medication use, laboratory data, and outcomes, were retrospectively collected from the Integrated Medical Database of NTUH (NTUH-iMD) and National Health Insurance Research Database (NHIRD). This study was approved by the institutional review board of NTUH.

### Participant selection

Adult patients aged over 20 yr who visited the ED with neurological symptoms due to suspected ischemic events and underwent carotid ultrasonography within 7 days after the ED visit were included. Neurological symptoms included hemiplegia, aphasia, facial palsy, central vertigo, conscious disturbance, or other focal neurological deficits. The neurologist will be consulted, and a carotid ultrasound will be arranged if an ischemic event was suspected.

### Assessment of patient data, carotid ultrasonography results and CPS, and outcomes

**Patient data.** The following information were obtained: demographic and clinical characteristics (age, sex, body mass index [BMI], and cigarette smoking), previous comorbidities (diabetes mellitus, hypertension, chronic kidney disease, CAD, and atrial fibrillation on electrocardiogram), medication history (oral hypoglycemic agents, insulin, antihypertensive drugs, statins, antiplatelets, and anticoagulants), BMI, and laboratory data (white blood cell [WBC] count, hemoglobin, platelet [PLT], creatinine, hemoglobin A1c, high-density lipoprotein [HDL], low-density lipoprotein, triglyceride, and total cholesterol). Based on the World Health Organization classification, BMI was categorized as follows: underweight in <18.5 kg/m$^2$; normal in 18.5–25.0 kg/m$^2$; pre-obesity in 25.0–30.0 kg/m$^2$; and obesity in ≥30 kg/m$^2$. Cigarette use was classified as follows: non-smoker, ex-smoker, and current smoker. BMI and laboratory examination results obtained during ED visits or at the nearest timing to ED visits were adopted.

**Carotid ultrasonography findings and CPS.** This study performed carotid ultrasonography within 7 days after the ED visit. HP (4500) or *Philips (iE33* or Affinity 70) ultrasound systems, equipped with a 7.5-MHz real-time B-mode scanner and a 5.6-MHz pulsed Doppler mode scanner, were used for the evaluation. The examination included the longitudinal and transverse view assessments of the bilateral carotid arteries. Experienced technicians performed carotid ultrasonography, and experienced neurologists evaluated stenotic severity. The bilateral common carotid arteries (CCAs), carotid bulbs, internal carotid arteries (ICAs), and external carotid arteries were evaluated. The degree of carotid stenosis was assessed using real-time B-mode ultrasonography images and the velocity criteria. The degree of carotid stenosis was investigated using B-mode ultrasonography following the North American Symptomatic Carotid Endarterectomy Trial criteria [25]. Carotid stenosis of ≥50%, which was considered significant carotid stenosis, was defined as a peak systolic velocity of ≥1.25 m/s and a peak systolic velocity between the stenotic and pre-stenotic segments of ≥2 m/s [26]. The scores of the eight carotid segments, which were as follows: 0, normal; 1, <30% stenosis; 2, 30%–49% stenosis; 3, 50%–69% stenosis; 4, 70%–99% stenosis; and 5, total occlusion, were added to determine the CPS [27]. Moreover, patients were determined for previously stent insertion. A CPS of

four was assigned in this case. Vertebrobasilar insufficiency (VBI) was defined as a bilateral vertebral artery flow of <100 mL/min [28].

**Outcomes.** All patients were followed up for 1 yr, and the mortality rate and cause of death were recorded based on the medical record collected from the NTUH-iMD database and the NHIRD Death Registry dataset. NHIRD was derived from the Bureau of National Health Insurance to enhance research on health care in Taiwan since 1995, which covers over 99% of Taiwan residents [29]. The cause of mortality was further categorized into CV-related, including cerebral vascular accident (International Classification of Diseases [ICD]-10 code: I60–I69) and cardiac-related and coronary artery disease-related (ICD-10 code: I01–I02.0, I05–I09, I20–I25, I27, and I30–I52), and non-CV-related. Further, the 1 yr, 180-day, and 90-day mortality rates, which represent short-term and long-term mortality, were recorded.

## Statistical analysis

Each continuous piece of data was assessed for normality using the Kolmogorov–Smirnov test. Dichotomous and categorical variables were presented as the absolute sample size (percentage), and continuous variables as mean (standard deviation [SD]) if normally distributed or median (interquartile range [IQR]) if not normally distributed. Using the Pearson chi-square test or the Fisher's exact test, categorical and nominal variables were compared. Normally distributed continuous variables were compared using the student $t$-test or the non-parametric analysis of variance and the Mann–Whitney U test if not normally distributed. The effects of significant carotid stenosis and CPS on 1-yr mortality were determined using the Cox proportional hazards model. All clinically important potential confounders, including demographic and clinical characteristics, previous comorbidities, use of antihypertensive drugs, insulin, statin, antiplatelets or anticoagulants, and laboratory data were adjusted using the Cox regression model with the forced entry method. Using CPS, the unadjusted Kaplan–Meier survival curve was depicted and categorized into four groups according to quartiles. Moreover, the association between CPS and other carotid ultrasonography results as well as 1-yr, 180-day, and 90-day mortality was investigated using the multivariate logistic regression model with all clinically important confounders adjusted using the forced entry method. Then, they were presented as adjusted odds ratio (aOR) and 95% confidence interval (CI). The regression model discrimination was evaluated using the area under the receiver operating characteristic (ROC) curve for each outcome. The model fit was assessed using the Hosmer–Lemeshow goodness-of-fit test. The ROC curve and the Youden Index were used to determine the best CPS cutoff value for predicting 1-yr all-cause and CV-related mortality. Statistical analysis was performed using the Statistical Package for the Social Sciences software version 26.0 (IBM, Armonk, NY, the USA). All tests were two-sided, and a $p$-value of <0.05 was considered statistically significant.

## Results

### Characteristics of participants

A total of 10,723 patients visited the ED with acute neurological deficits from 2009 to 2018. Among them, 7,961 underwent complete carotid ultrasonography of the eight carotid segments, including the carotid bulb, CCA, ICA, and external carotid arteries. The final cohort had 746 (9.4%) patients who died within a 1-yr follow-up, including 247 (3.1%) with CV-related mortality. Fig 1 shows the detailed flow diagram.

Demographic and clinical characteristics of patients were compared with 1-yr mortality as shown in Table 1. The median age of all participants was 69.0 yr, and approximately 58.7% were males. Patients who died within 1 yr were significantly older (median age: 77.0 vs. 69.0 yr; $p < 0.001$) and were less frequently obese and smokers. CAD (14.7% vs. 9.1%, $p < 0.001$)

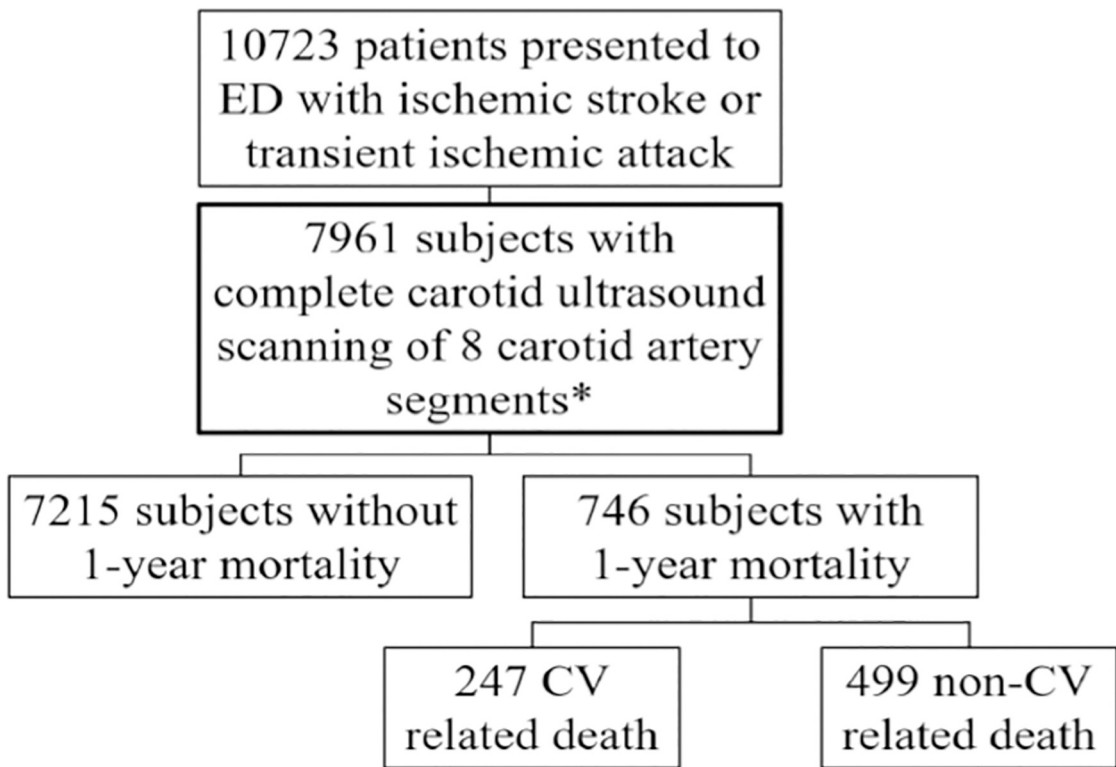

**Fig 1. Flow chart of included patients.** CV = cardiovascular, ED = emergency department. *Include carotid bulb, common carotid artery, internal carotid artery and external carotid artery of both sides.

and atrial fibrillation (34.7% vs. 22.2%; $p < 0.001$) history were more common in the mortality group. Patients with 1-yr mortality frequently used more antihypertensive drugs (75.3% vs. 68.9%; $p < 0.001$) and less statins (30.7% vs. 40.8%; $p < 0.001$) and antiplatelets (58.4% vs. 68.0%; $p < 0.001$). A higher median WBC count and creatinine level and lower hemoglobin, PLT, HDL, low-density lipoprotein, triglyceride, and total cholesterol levels were determined in the mortality group. Patients with 1-yr mortality presented more significant stenosis in the carotid bulb (2.9% vs. 0.9%, $p < 0.001$), CCA (1.6% vs. 0.7%, $p = 0.006$), and ICA (15.0% vs. 7.2%, $p < 0.001$) than those without 1-yr mortality. Patients with 1-yr mortality had a higher median CPS than those without 1-yr mortality (6.0 vs. 4.0, $p < 0.001$). S1 Table in S1 File shows the number and percentage of carotid stenosis at varying degrees in each segment.

## Mortality

Cox regression analysis results are shown in Table 2. Significant carotid stenosis, either in the carotid bulb, CCA, or ICA, was not associated with a significantly higher all-cause, CV, or non-CV-related mortality. A higher CPS was associated with a greater 1-yr all-cause mortality (aHR = 1.08, 95% CI = 1.03–1.13, $p = 0.001$) and CV-related mortality (aHR = 1.13, 95% CI = 1.04–1.22, $p = 0.002$). The VBI was also correlated with a higher CV-related mortality rate, with an aHR of 2.05 (95% CI = 1.15–3.62, $p = 0.014$). A higher mortality rate was associated with CAD and CV disease and the all-cause mortality with lower PLT and HDL levels. Patients treated with antihypertensive drugs were less likely to have CV-related mortality (aHR = 0.45, 95% CI = 0.22–0.93, $p = 0.032$). Meanwhile, the use of anti-platelet drugs was correlated with lower mortality (regardless of all-cause, CV, or non-CV). Fig 2 shows the

**Table 1. Basic characteristics of included patients.**

| Variables [a] | Total (n = 7961) | Survival within 1 year (n = 7215) | Mortality within 1 year (n = 746) | p |
|---|---|---|---|---|
| Age | 69.0 (20.0) | 69.0 (19.0) | 77.0 (20.0) | <**0.001** |
| Male | 4676 (58.7) | 4241 (58.8) | 435 (58.3) | 0.804 |
| BMI | | | | <**0.001** |
| Underweight | 255 (3.8) | 204 (3.4) | 51 (8.5) | |
| Normal | 3475 (52.2) | 3133 (51.7) | 342 (57.1) | |
| Pre-obesity | 2270 (34.1) | 2102 (34.7) | 168 (28.0) | |
| Obesity | 655 (9.8) | 617 (10.2) | 38 (6.3) | |
| Smoking | | | | **0.039** |
| Non-smoker | 6116 (88.4) | 5723 (88.2) | 393 (91.8) | |
| Ex-smoke | 386 (5.6) | 373 (5.7) | 13 (3.0) | |
| Smoker | 416 (6.0) | 394 (6.1) | 22 (5.1) | |
| DM | 2804 (35.2) | 2532 (35.1) | 272 (36.5) | 0.457 |
| HTN | 5162 (64.8) | 4682 (64.9) | 480 (64.3) | 0.765 |
| CKD | 1095 (13.8) | 1004 (13.9) | 91 (12.2) | 0.195 |
| CAD | 764 (9.6) | 654 (9.1) | 110 (14.7) | <**0.001** |
| Afib | 1858 (23.3) | 1599 (22.2) | 259 (34.7) | <**0.001** |
| OHA | 2282 (28.7) | 2043 (28.3) | 239 (32.0) | **0.032** |
| Insulin | 580 (7.3) | 522 (7.2) | 58 (7.8) | 0.589 |
| Anti-hypertensive | 5532 (69.5) | 4970 (68.9) | 562 (75.3) | <**0.001** |
| Statin | 3174 (39.9) | 2945 (40.8) | 229 (30.7) | <**0.001** |
| Anti-platelet | 5343 (67.1) | 4907 (68.0) | 436 (58.4) | <**0.001** |
| Anti-coagulant | 895 (11.2) | 821 (11.4) | 74 (9.9) | 0.230 |
| WBC | 7.1 (3.1) | 7.0 (2.9) | 10.2 (7.5) | <**0.001** |
| Hb | 12.9 (3.3) | 13.1 (3.3) | 10.2 (3.2) | <**0.001** |
| PLT | 220.0 (94.0) | 223.0 (91.0) | 191.0 (140.0) | <**0.001** |
| CRE | 1.0 (0.5) | 0.9 (0.4) | 1.1 (1.4) | <**0.001** |
| HbA1c | 5.9 (1.1) | 5.9 (1.0) | 5.9 (1.4) | 0.953 |
| HDL | 43.0 (17.0) | 43.0 (16.0) | 38.0 (16.0) | <**0.001** |
| LDL | 93.0 (41.0) | 93.0 (40.0) | 90.0 (43.0) | **0.005** |
| TG | 109.0 (71.0) | 110.0 (72.0) | 100.0 (69.0) | <**0.001** |
| Total cholesterol | 163.0 (50.0) | 164.0 (50.0) | 151.0 (56.0) | <**0.001** |
| Significant carotid stenosis [b] | | | | |
| Carotid bulb | 87 (1.1) | 65 (0.9) | 22 (2.9) | <**0.001** |
| CCA | 61 (0.8) | 49 (0.7) | 12 (1.6) | **0.006** |
| ICA | 634 (8.0) | 622 (7.2) | 112 (15.0) | <**0.001** |
| Any previous stenting | 30 (0.4) | 27 (0.4) | 3 (0.4) | 0.757 |
| Carotid plaque score | 4.0 (5.0) | 4.0 (4.0) | 6.0 (5.0) | <**0.001** |
| Vertebrobasilar insufficiency | 736 (9.3) | 631 (8.8) | 104 (14.4) | <**0.001** |

Afib = atrial fibrillation, BMI = body mass index, CAD = coronary artery disease, CCA = common carotid artery, CKD = chronic kidney disease, CRE = creatinine, CVA = cerebral vascular accident, DM = diabetes mellitus, HB = hemoglobin, HDL = High–density lipoprotein, HTN = hypertension, ICA = internal carotid artery, LDL = low–density lipoprotein, OHA = oral hypoglycemic medications, PLT = platelet, TG = triglyceride, WBC = white blood cell

[a] Dichotomous and categorical variables were reported as absolute sample size (percentages), whereas continuous variables were reported as median (IQR)

[b] >50% stenosis of either side

**Table 2. Cox regression of mortality within 1–year follow up.**

| | All-cause mortality | | CV related mortality | | Non-CV related mortality | |
|---|---|---|---|---|---|---|
| | adjusted HR (95% CI) | *p* | adjusted HR (95% CI) | *p* | adjusted HR (95% CI) | *p* |
| Age | 1.00 (0.98–1.01) | 0.703 | 0.99 (0.97–1.02) | 0.438 | 1.00 (0.98–1.02) | 0.989 |
| Male | 0.78 (0.57–1.07) | 0.128 | 0.73 (0.42–1.26) | 0.253 | 0.82 (0.55–1.22) | 0.321 |
| BMI | | | | | | |
| Underweight | 1.09 (0.59–2.00) | 0.788 | 1.00 (0.37–2.73) | 0.998 | 1.13 (0.52–2.46) | 0.761 |
| Normal | Reference | | Reference | | Reference | |
| Pre-obesity | 0.82 (0.59–1.16) | 0.265 | 0.78 (0.43–1.41) | 0.417 | 0.85 (0.56–1.31) | 0.459 |
| Obesity | 0.99 (0.58–1.69) | 0.955 | 0.94 (0.38–2.36) | 0.895 | 1.11 (0.57–2.17) | 0.764 |
| Smoking | | | | | | |
| Non-smoker | Reference | | Reference | | Reference | |
| Ex-smoke | 1.17 (0.55–2.48) | 0.692 | 1.67 (0.55–5.10) | 0.369 | 0.87 (0.31–2.49) | 0.797 |
| Smoker | 0.84 (0.42–1.71) | 0.637 | 0.25 (0.03–1.97) | 0.190 | 1.12 (0.51–2.48) | 0.784 |
| DM | 1.11 (0.71–1.74) | 0.642 | 1.17 (0.53–2.56) | 0.701 | 1.22 (0.70–2.12) | 0.486 |
| HTN | 0.74 (0.50–1.09) | 0.128 | 0.81 (0.40–1.62) | 0.552 | 0.69 (0.42–1.12) | 0.132 |
| CKD | 0.91 (0.63–1.31) | 0.611 | 1.03 (0.54–1.97) | 0.935 | 0.86 (0.55–1.35) | 0.519 |
| CAD | 1.32 (0.88–1.97) | 0.178 | 2.50 (1.29–4.85) | **0.007** | 0.83 (0.48–1.43) | 0.506 |
| Afib | 0.86 (0.62–1.18) | 0.344 | 1.40 (0.80–2.47) | 0.241 | 0.64 (0.43–0.95) | **0.026** |
| OHA | 0.90 (0.56–1.43) | 0.644 | 1.01 (0.45–2.28) | 0.974 | 0.82 (0.46–1.45) | 0.489 |
| Insulin | 0.66 (0.39–1.11) | 0.117 | 0.51 (0.19–1.37) | 0.180 | 0.74 (0.39–1.40) | 0.356 |
| Anti-hypertensive | 0.72 (0.46–1.12) | 0.144 | 0.45 (0.22–0.93) | **0.032** | 0.92 (0.51–1.65) | 0.778 |
| Statin | 0.84 (0.61–1.16) | 0.285 | 0.77 (0.44–1.36) | 0.371 | 0.86 (0.58–1.28) | 0.470 |
| Anti-platelet | 0.52 (0.37–0.74) | **<0.001** | 0.44 (0.25–0.78) | **0.005** | 0.56 (0.36–0.86) | **0.009** |
| Anti-coagulant | 0.93 (0.64–1.37) | 0.726 | 1.05 (0.57–1.92) | 0.876 | 0.91 (0.55–1.50) | 0.710 |
| WBC | 1.03 (1.01–1.04) | **<0.001** | 1.02 (0.99–1.06) | 0.239 | 1.03 (1.01–1.04) | **0.001** |
| Hb | 0.95 (0.89–1.02) | 0.180 | 1.10 (0.98–1.24) | 0.104 | 0.87 (0.79–0.96) | **0.005** |
| PLT | 0.99 (0.99–1.00) | **0.049** | 1.00 (1.00–1.00) | 0.622 | 0.99 (0.99–1.00) | **0.048** |
| CRE | 0.93 (0.84–1.02) | 0.140 | 0.89 (0.73–1.09) | 0.272 | 0.94 (0.84–1.05) | 0.247 |
| HbA1C | 0.96 (0.83–1.12) | 0.607 | 0.96 (0.73–1.27) | 0.773 | 0.99 (0.83–1.19) | 0.911 |
| HDL | 0.99 (0.98–0.99) | **0.036** | 1.00 (0.98–1.02) | 0.851 | 0.98 (0.96–0.99) | **0.006** |
| LDL | 1.00 (0.99–1.01) | 0.728 | 1.00 (0.99–1.01) | 0.709 | 1.00 (0.99–1.01) | 0.528 |
| TG | 1.00 (1.00–1.00) | 0.428 | 1.00 (1.00–1.00) | 0.124 | 1.00 (1.00–1.00) | 0.398 |
| Total cholesterol | 1.00 (1.00–1.01) | 0.868 | 1.00 (0.99–1.01) | 0.506 | 1.00 (1.00–1.01) | 0.338 |
| Significant carotid stenosis [a] | | | | | | |
| Carotid bulb | 0.84 (0.29–2.41) | 0.747 | NA | | 1.70 (0.56–5.14) | 0.347 |
| CCA | 0.46 (0.10–2.18) | 0.328 | 1.58 (0.19–13.13) | 0.670 | 0.30 (0.04–2.53) | 0.268 |
| ICA | 0.67 (0.38–1.16) | 0.149 | 0.58 (0.23–1.43) | 0.236 | 0.70 (0.35–1.43) | 0.332 |
| **Carotid plaque score** | **1.08 (1.03–1.13)** | **0.001** | **1.13 (1.04–1.22)** | **0.002** | 1.05 (0.99–1.12) | 0.098 |
| Any previous stenting | NA | | NA | | NA | |
| Vertebrobasilar insufficiency | 0.99 (0.64–1.53) | 0.965 | 2.05 (1.15–3.62) | **0.014** | 0.45 (0.21–0.98) | **0.044** |

Afib = atrial fibrillation, BMI = body mass index, CAD = coronary artery disease, CCA = common carotid artery, CI = confidence interval, CKD = chronic kidney disease, CRE = creatinine, CV = cardiovascular, DM = diabetes mellitus, HB = hemoglobin, HDL = High–density lipoprotein, HR = hazard ratio, HTN = hypertension, ICA = internal carotid artery, LDL = low–density lipoprotein, OHA = oral hypoglycemic medications, PLT = platelet, TG = triglyceride, WBC = white blood cell

[a] >50% stenosis of either side

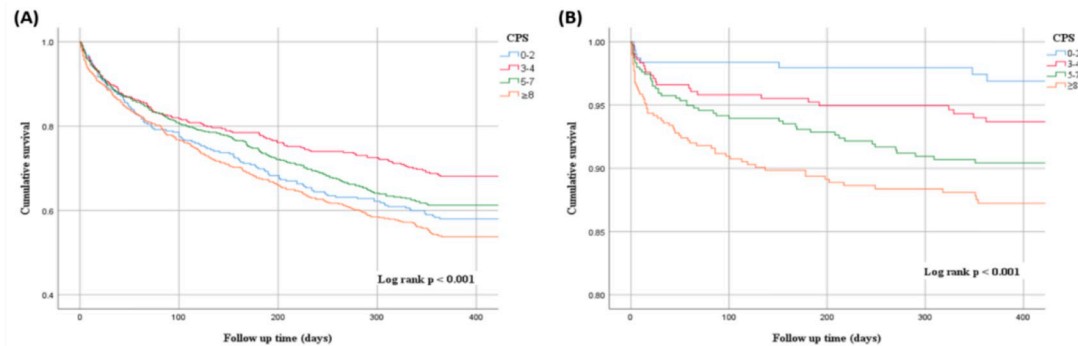

**Fig 2. Cumulative 1–year survival between different carotid plaque score (CPS) groups.** (A) all–cause mortality (B) CV–related mortality.

cumulative 1-yr survival among patients with a CPS of 0–2, 3–4, 5–7, and ≥8. Patients with higher CPS scores had a lower cumulative survival for all-cause and CV-related mortality (log-rank $p < 0.001$).

The multivariate logistic regression model was used to analyze the association between CPS and 1-yr, 180-day, and 90-day mortality as shown in Table 3. A higher CPS was correlated with a greater risk of 1-yr all-cause and CV-related mortality, 180-day all-cause and CV-related and non-CV-related mortality, and 90-day all-cause and CV-related mortality. The Hosmer–Lemeshow test results showed a good fit in most models. The area under the ROC curve of all logistic regression models reached >0.8, which indicated good mortality discrimination and predictive accuracy (S1 Fig in S1 File).

The ROC curve and Youden Index were further used to determine the best CPS cutoff value for predicting 1-yr all-cause and CV-related mortality (S2 Fig in S1 File). The best cutoff

**Table 3. Association of carotid plaque score and mortality [a].**

|  | *n* (%) | Carotid plaque score | *p* |
|---|---|---|---|
| Multivariable Cox regression |  |  |  |
| All-cause mortality, HR (95%CI) | 746 (9.4) | 1.08 (1.03–1.13) | **0.001** |
| CV related mortality, HR (95%CI) | 247 (3.1) | 1.13 (1.04–1.22) | **0.002** |
| Non-CV related mortality, HR (95%CI) | 499 (6.3) | 1.05 (0.99–1.12) | 0.098 |
| Multivariable logistic regression |  |  |  |
| 1-year mortality, aOR (95%CI) | 746 (9.4) | 1.10 (1.04–1.16) | **0.001** |
| 1-year CV related mortality, aOR (95%CI) | 247 (3.1) | 1.16 (1.06–1.26) | **0.001** |
| 1-year Non-CV related mortality, aOR (95%CI) | 499 (6.3) | 1.05 (0.98–1.13) | 0.165 |
| 180-day mortality, aOR (95%CI) | 514 (6.5) | 1.16 (1.09–1.24) | **<0.001** |
| 180-day CV related mortality, aOR (95%CI) | 187 (2.3) | 1.24 (1.13–1.37) | **<0.001** |
| 180-day non-CV related mortality, aOR (95%CI) | 327 (4.1) | 1.10 (1.01–1.20) | **0.023** |
| 90-day mortality, aOR (95%CI) | 372 (4.7) | 1.15 (1.07–1.24) | **<0.001** |
| 90-day CV related mortality, aOR (95%CI) | 155 (1.9) | 1.24 (.11–1.38) | **<0.001** |
| 90-day non-CV related mortality, aOR (95%CI) | 217 (2.7) | 1.08 (0.97–1.19) | 0.160 |

aOR = adjusted odds ratio, CI = confidence interval, CV = cardiovascular, HR = hazard ratio, NA = not available, STEMI = ST–elevated myocardial infarction

[a] All regression models were adjusted for age, sex, body mass index, cigarette use, past comorbidities, use of antihypertensives drugs, insulin, statins, antiplatelets or anticoagulants, and laboratory data

values with the maximum Youden Index were ≥6 in all-cause mortality and ≥5 in CV-related mortality.

## Discussion

This study investigated the association of several carotid ultrasound parameters and mortality in the 10-yr cohort of patients who presented to the ED with neurological deficit. Results had shown that higher CPS was associated with a greater risk of 1-yr all-cause and CV-related mortality. CPS, rather than stenotic severity in a single carotid segment, was more significantly correlated with 1-yr mortality.

This study had several strengths. First, this study comprehensively investigated the association between carotid ultrasound parameters, including carotid stenosis, CPS, and mortality among patients with suspected cerebral ischemic events. The results can be an important guide for etiological surveys and risk stratifications. Second, our research had a large cohort (over 7,000 patients), which can reflect real-world conditions and add statistical power. To our best knowledge, this study had the largest cohort and longest duration compared with previous studies. Third, we adjusted for several important confounders, including demographic characteristics, previous comorbidities, medication use, and laboratory data. The outcomes of our study, the mortality and causes of death, were collected from NHIRD, which has very high coverage of all residents in Taiwan, limiting the chance of potential bias caused by patients who were not followed up with. The previous study also had shown a good accuracy and correctness of this dataset [29].

In contrast to previous studies, our study failed to show a significant correlation between a carotid bulb, CCA, or ICA stenosis and 1-yr mortality [20, 21]. However, the only parameter of carotid ultrasound that was investigated in those studies was the degree of carotid stenosis [20, 21]. Our study revealed that CPS, which indicates total plaque burden, had shown a stronger risk factor of mortality than significant carotid stenosis in a single segment. Our result may be well explained because fatal CV diseases will lead to higher mortality. After all, studies had shown an association between higher CPS and a higher risk of CAD and other major CV events [16–18, 23].

This cohort study revealed that most patients presented significant stenosis in ICA, with a rate of approximately 8%, which is slightly lower than the rate of 12.5% in patients with TIA or ischemic stroke in the previous study [30]. These patients were considered symptomatic and with significant carotid stenosis, which required timely interventions, such as carotid endarterectomy; thus, a survey of carotid stenosis by ultrasound, especially the ICA, was essential in ED to ensure better outcomes [31].

We presented a good prediction model for short-term and long-term mortality among patients with neurological symptoms in 1 yr with CPS and carotid ultrasonography results. The current study reached a c-statistics of over 0.8 for predicting 1-yr, 180-day, and 90-day all-cause and CV-related mortality (S1 Fig in S1 File). The current study established a prediction model for patients with suspected TIA or ischemic stroke in the ED by adding the carotid ultrasound results to other previously known risk factors, such as age, pre-existing comorbidities, and laboratory data. In our study, a CPS of ≥6 predicted all-cause mortalities, whereas a CPS of ≥5 may indicate higher CV-related mortality. The method of calculating the CPS in our study had not been verified; however, patients with CPS higher than the cutoff value indicated a higher risk of negative outcomes, which require further medical treatment or interventions.

Our research revealed a possible association of other risk factors with a greater mortality rate as higher WBC count and a lower PLT level. The inflammation indicates important

pathogenesis in atherosclerosis [32]. Previous studies have shown a correlation between a lower PLT level-to-WBC count ratio, which indicated a lower PLT level and a higher WBC count, as well as stroke severity and mortality [33, 34]. The use of antiplatelets and antihypertensive drugs was associated with a lower risk of CV-related mortality. Other medications for metabolic syndromes may only be beneficial for longer term outcomes.

This study has several limitations. First, all retrospective cohort studies have an inherent bias. Some factors and variables, such as initial ED diagnosis and the stroke percentage and its subtypes or severity (such as the National Institutes of Health Stroke Scale), were not available in this study. Also, only 71.7% (7,691 out of 10,723) patients with acute neurological deficit in ED received carotid artery exams. Excluding patients without carotid ultrasound exams may lead to a selection bias. Second, carotid interventions, such as stenting and endarterectomy after carotid ultrasonography, were not assessed. Finally, the CPS calculation and VBI definition were still not elucidated. Therefore, further studies are required to verify these results.

This research supported the utilization of carotid ultrasonography and CPS in predicting mortality among patients with neurological deficits in the ED. However, further studies should be conducted to determine whether CPS can be used as a guide for future interventions. The potential benefit of lifestyle modification, medication use (such as using stronger anti-platelet and dual anti-platelet therapy), or even interventional therapies in such patients remained unknown.

## Conclusion

Patients with ischemic neurological symptoms with ED visits have a higher CPS, which is correlated with a greater risk of 1-yr mortality, particularly CV-related. Such patients should undergo carotid ultrasonography and a CPS calculation for risk stratification. However, further studies must be conducted to assess the use of CPS as a guide in different interventions or medical decision-making among such patients.

## Supporting information

**S1 Checklist. STROBE statement—Checklist of items that should be included in reports of observational studies.**
(DOCX)

**S1 File.**
(DOCX)

## Acknowledgments

We thank the staff of Department of Medical Research, National Taiwan University Hospital Hsin-Chu Branch for their assistance in data cleansing and statistical analysis. We also thank the English editors of www.enago.tw for helping with language editing.

## Author Contributions

**Conceptualization:** Chih-Wei Sung, Jiann-Shing Jeng, Sung-Chun Tang, Edward Pei-Chuan Huang.

**Data curation:** Chi-Hsin Chen, Chih-Wei Sung, Cheng-Yi Fan.

**Formal analysis:** Chi-Hsin Chen.

**Funding acquisition:** Edward Pei-Chuan Huang.

**Methodology:** Chi-Hsin Chen, Chih-Wei Sung, Jia-How Chang, Jiun-Wei Chen, Sung-Chun Tang, Edward Pei-Chuan Huang.

**Resources:** Jiann-Shing Jeng, Sung-Chun Tang.

**Supervision:** Jiann-Shing Jeng, Jia-How Chang, Jiun-Wei Chen, Sung-Chun Tang, Edward Pei-Chuan Huang.

**Validation:** Chih-Wei Sung, Cheng-Yi Fan, Jia-How Chang, Jiun-Wei Chen.

**Visualization:** Cheng-Yi Fan.

**Writing – original draft:** Chi-Hsin Chen.

**Writing – review & editing:** Sung-Chun Tang, Edward Pei-Chuan Huang.

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
