## [Decision Letter · Decision Letter 0]

18 Aug 2022

PONE-D-22-19297Utility of carotid ultrasound on prediction of 1-year mortality in emergency department patients with neurological deficits: A 10-year population-based cohort studyPLOS ONE

Dear Dr. Pei-Chuan Huang,

Thank you for submitting your manuscript to PLOS ONE. After careful consideration, we feel that it has merit but does not fully meet PLOS ONE’s publication criteria as it currently stands. Therefore, we invite you to submit a revised version of the manuscript that addresses the points raised during the review process.

We look forward to receiving your revised manuscript.

Kind regards,

Seung-Hwa Lee

Academic Editor

PLOS ONE

Journal Requirements:

Reviewers' comments:

Reviewer's Responses to Questions

**Comments to the Author**

1. Is the manuscript technically sound, and do the data support the conclusions?

Reviewer #1: Yes

Reviewer #2: Yes

Reviewer #3: Partly

2. Has the statistical analysis been performed appropriately and rigorously? 

Reviewer #1: Yes

Reviewer #2: Yes

Reviewer #3: Yes

3. Have the authors made all data underlying the findings in their manuscript fully available?

Reviewer #1: Yes

Reviewer #2: No

Reviewer #3: No

4. Is the manuscript presented in an intelligible fashion and written in standard English?

Reviewer #1: Yes

Reviewer #2: Yes

Reviewer #3: Yes

5. Review Comments to the Author

Reviewer #1: dear author

The submitted text is suitable for publication in the journal, but in the introduction section, you must clearly state the purpose of your study, and in the abstract section, be sure to compare the results of your study with several similar studies and state the superiority of your study over other researches. .

Thanks

Reviewer #2: From my point of view, it is necessary to clarify several points:

- Do all patients with suspected stroke undergo a carotid ultrasound? Is it done in all cases? This needs to be clarified. If this were not the case, a selection bias could have been incurred, if only the most severe patients underwent the technique. This should be described as a limitation.

- Line 191: "The Kolmogorov–Smirnov test was used to assess all continuous data". This has been already mentioned in the Statistical Analysis section. Delete.

- Results: Patients who died within 1 year were less frequently obese and smokers. Describe this in Results section since this information is somehow unespected.

- Revise lines 198-200. LDL was not lower in the mortality group as it is shown in Table 1.

- Maybe depict in bold significant results in all tables to ease understand the data.

- I understand that Table 2 is univariate, is it?. If this is the case, this has to be clearly state in the text and in the table title or footnote.

- Why is logistic regression used for table 3? Doesn't it seem more logical to perform Cox regression again to relate the presence of plaque (independent variable) with mortality, etc.?

The variables for which it was adjusted must be described exactly, it does not apply to talk about 'medications'. What criteria was used to select the confounders? clinical, statistical criteria?, which ones?, how?.

Reviewer #3: The correlation between CPS and mortality has been already described in the literature (J Atheroscler Thromb. 2018 Jan 1; 25(1): 55–64, Hypertens Res 2013; 36, 902–909), as well as the association between CPS and the risk of CVD (Journal of Cardiology (2008) 51, 25–32).

Furthermore, the 2016 ESC Guidelines on Cardiovascular Disease Prevention have included plaque detection as a modifier in cardiovascular risk assessment (class IIb, level B) after the initial assessment has been performed using established risk scores, in symptomatic patients.

The novelty of the work is there limited.

An English revision should also be performed.

6. PLOS authors have the option to publish the peer review history of their article (what does this mean?). If published, this will include your full peer review and any attached files.

Reviewer #1: No

Reviewer #2: No

Reviewer #3: No

---

## [Author Response · Author response to Decision Letter 0]

3 Sep 2022

Response to Reviewers’ Comments 

Manuscript Number: PONE-D-22-19297

Title: Utility of carotid ultrasound on prediction of 1-year mortality in emergency department patients with neurological deficits: A 10-year population-based cohort study

Dear editors and reviewers: 

Thanks for the insightful comments from the reviewers. We have revised our manuscript extensively according to all reviewers’ suggestion including the major concern of novelty and clinical significance of our study from the third reviewer and several concerns about statistical method from the second reviewer. Also, as per the advice of the third reviewer, our manuscript had been resent for English editing again. The detailed point-by-point responses are as follows. We hope the manuscript is now acceptable to PLoS One.

Yours truly,

Edward Pei-Chuan Huang, M.D., M.S.

Sung-Chun Tang, M.D., Ph.D.

Corresponding Authors 

Reviewer #1:

dear author

The submitted text is suitable for publication in the journal, but in the introduction section, you must clearly state the purpose of your study, and in the abstract section, be sure to compare the results of your study with several similar studies and state the superiority of your study over other researches. 

Thanks

Response from the authors: 

Thanks for your insightful suggestion. We had added statements to emphasize the objective of our study in the last paragraph of Introduction

Also, we added some important strengths which may be superior to previous studies in the Conclusion section of Abstract.

Changes made (page numbered on marked copy): <Page 7, Line 107-110 >

The objective is to investigate the utility and the optimal parameter of carotid ultrasound on prediction of mortality. The results of the current study may be an important guide for risk stratifications in ED practice.

Changes made (page numbered on marked copy): <Page 4, Line 55-60>

We comprehensively investigated the utility of carotid ultrasound parameters on predicting mortality in this 10-yr population-based cohort, which included over 7,000 patients with acute neurological deficits presented to the ED. The result showed that CPS could be used as risk stratification tools for 1-yr all-cause and CV mortality.

Reviewer #2:

From my point of view, it is necessary to clarify several points:

- Do all patients with suspected stroke undergo a carotid ultrasound? Is it done in all cases? This needs to be clarified. If this were not the case, a selection bias could have been incurred, if only the most severe patients underwent the technique. This should be described as a limitation.

Response from the authors: 

Thanks for reminding. Carotid ultrasound was not performed in all cases. In the “Result-Characteristics of participants” section, we had stated that 7,961 out of 10,723 patients with suspected stroke received carotid ultrasound exam. As your opinion, this indeed may present selection bias. We had added it in the limitation section. 

Changes made (page numbered on marked copy): <Page 25, Line 317-319>

Also, only 71.7% (7,691 out of 10,723) patients with acute neurological deficit in ED received carotid artery exams. Excluding patients without carotid ultrasound exams may lead to a selection bias.

- Line 191: "The Kolmogorov–Smirnov test was used to assess all continuous data". This has been already mentioned in the Statistical Analysis section. Delete.

Response from the authors: 

Thanks for your advice, the redundant sentence had been deleted. 

Changes made (page numbered on marked copy): 

The redundant sentence had been deleted.

- Results: Patients who died within 1 year were less frequently obese and smokers. Describe this in Results section since this information is somehow unexpected.

Response from the authors: 

Thanks for your suggestion, we have incorporated this description into this paragraph.

Changes made (page numbered on marked copy): <Page 13, Line 216-218>

Patients who died within 1 yr were significantly older (median age: 77.0 vs. 69.0 yr; p < 0.001) and were less frequently obese and smokers.

- Revise lines 198-200. LDL was not lower in the mortality group as it is shown in Table 1.

Response from the authors: 

Thanks for your professional and meticulous review. We wonder if there is a misunderstanding. In Table 1, the LDL was lower in the mortality group (median 90 vs 93, p=0.005). This was also an unexpected result; however, no significant association of LDL and mortality was reported in the multivariable Cox regression (Table 2).

Changes made (page numbered on marked copy):

No changes were made

- Maybe depict in bold significant results in all tables to ease understand the data.

Response from the authors: 

Thanks for your suggestion, we had changed the p value with significant results to bold type.

Changes made (page numbered on marked copy): 

Table 1-3

- I understand that Table 2 is univariate, is it?. If this is the case, this has to be clearly state in the text and in the table title or footnote.

Response from the authors: 

We apologized for causing misunderstanding. The results in Table 2 were conducted by multivariable Cox regression. We thought this misunderstanding resulted from the presenting the column name as “Hazard ratio” instead of adjusted hazard ratio. We had changed the column name from Hazard ratio to adjusted HR, hope it could be more understandable. 

Changes made (page numbered on marked copy): <Table 2 >

Column name “Hazard ratio” to “adjusted HR”.

- Why is logistic regression used for table 3? Doesn't it seem more logical to perform Cox regression again to relate the presence of plaque (independent variable) with mortality, etc.?

Response from the authors: 

Thanks for your insightful advice. The reason for performing logistic regression was that we tried to investigate the utility of carotid ultrasound parameters on short to longer-term of mortality (90d, 180d, 1yr). This study further demonstrated the predictive accuracy of the logistic regression model for 90-day, 180-day, and 1-year all-cause and CV mortality using area under ROC curve (AUROC), as presented in Supplementary Figure 1 (The model showed good discrimination [AUROC>0.8] for all outcomes). None of the above can be fully presented through Cox regression. We also revised Table 3 by adding the statistical method to make it easy to read. Mentioned to use Cox regression for mortality analysis, we thought it’s also reasonable and insightful advice. In the multivariable cox regression model (Table 2), significant carotid stenosis (plaques causing a >50% stenosis, which should be more fatal compared to presence of plaque only), was not significant after adjusted in the multivariable Cox regression. This may indicate that CPS were a stronger indicator of mortality than other parameters. Therefore, we would like to focus on investigating the utility of carotid plaque score more. We sincerely hope that we do answer your in-depth questions and comments. 

Changes made (page numbered on marked copy): 

Table 3

The variables for which it was adjusted must be described exactly, it does not apply to talk about 'medications'. What criteria was used to select the confounders? clinical, statistical criteria?, which ones?, how?.

Response from the authors: 

Thanks for your comments. We had specified “medication use” to “use of antihypertensive drugs, insulin, statin, antiplatelets or anticoagulants.” Besides, we selected all confounders (as variables listed in Table 1) with clinical importance to multivariable regression models. The manuscript and table were revised to elaborate the analysis.

Changes made (page numbered on marked copy): 

Table 3

Changes made (page numbered on marked copy): <Page 11, Line 187-191>

All clinically important potential confounders, including demographic and clinical characteristics, previous comorbidities, use of antihypertensive drugs, insulin, statin, antiplatelets or anticoagulants, and laboratory data were adjusted using the Cox regression model with the forced entry method.

Changes made (page numbered on marked copy): <Page 11, Line 193-196>

Moreover, the association between CPS and other carotid ultrasonography results as well as 1-yr, 180-day, and 90-day mortality was investigated using the multivariate logistic regression model with all clinically important confounders adjusted using the forced entry method.

Reviewer #3:

The correlation between CPS and mortality has been already described in the literature (J Atheroscler Thromb. 2018 Jan 1; 25(1): 55–64, Hypertens Res 2013; 36, 902–909), as well as the association between CPS and the risk of CVD (Journal of Cardiology (2008) 51, 25–32).

Furthermore, the 2016 ESC Guidelines on Cardiovascular Disease Prevention have included plaque detection as a modifier in cardiovascular risk assessment (class IIb, level B) after the initial assessment has been performed using established risk scores, in symptomatic patients.

The novelty of the work is there limited.

An English revision should also be performed.

Response from the authors: 

Thanks for taking your precious time to review our articles and give us valuable comments. As you mentioned, several previous studies had investigated the topic of CPS. However, there were still some difference and strengths in our study demonstrates the importance and academic value of our research (A detail comparison table were presented in the following table). First, the study group were different. As written in the introduction section, the included subjects were elder patients aged ≥85 years in “J Atheroscler Thromb. 2018 Jan 1; 25(1): 55–6” and hypertensive patients in “Hypertens Res 2013; 36, 902–909”. In our study, we focused on patients with acute neurological deficits in ED. Besides, the results of previous studies showed divergence (the former with significant results and the latter only reported a trend of increased risk). Furthermore, our study included a 10-year and larger cohort (over 7,000 comparing to around 300 patients) and compared different carotid ultrasound parameters. All above may guide interpretations of carotid ultrasound results. 

Comparison of previous studies to our study

 Hirata.et.al1

Kawai.et.al2

Our study

Study group Japanese subjects aged ≥85 years without cardiovascular disease Hypertensive Japanese patients in the outpatient clinic ED Patients with neurological deficits in Taiwan

Number of included patients 347 356 7,961

Primary outcome 6-year CV mortality Long term mortality

(mean 6.4 years) 1-year all cause and CV mortality

Result High CPS correlated significantly with higher CV mortality. Higher CPS showed a non-significant trend of higher risk of mortality High CPS correlated significantly with higher CV and all-cause mortality

The suggestion of detecting plaque as CV risk stratification tool were based on a population-based study performed by O'Leary.et.al.3, 4 However, the study did not include CPS as a carotid ultrasound parameter ,and the optimal carotid ultrasound parameter for risk stratification was unknown. 

Finally, we apologized for causing you a misunderstanding that this manuscript was not reviewed by experienced editor whose first language is English. In fact, our manuscript was edited by English editors in www.Enago.com before submission. We will resend our manuscript again for more detailed editing. Thanks for your kind reminder. 

Changes made (page numbered on marked copy): 

The manuscript was re-edited by Crimson Interactive Inc., an editing brand of Enago, for language, grammar, structure, and content, from the aspect of fluency and nativity.

Reference

1. Hirata T, Arai Y, Takayama M, Abe Y, Ohkuma K, Takebayashi T. Carotid Plaque Score and Risk of Cardiovascular Mortality in the Oldest Old: Results from the TOOTH Study. J Atheroscler Thromb. 2018;25:55-64.

2. Kawai T, Ohishi M, Takeya Y, et al. Carotid plaque score and intima media thickness as predictors of stroke and mortality in hypertensive patients. Hypertens Res. 2013;36:902-909.

3. Piepoli MF, Hoes AW, Agewall S, et al. 2016 European Guidelines on cardiovascular disease prevention in clinical practice: The Sixth Joint Task Force of the European Society of Cardiology and Other Societies on Cardiovascular Disease Prevention in Clinical Practice (constituted by representatives of 10 societies and by invited experts)Developed with the special contribution of the European Association for Cardiovascular Prevention & Rehabilitation (EACPR). Eur Heart J. 2016;37:2315-2381.

4. O'Leary DH, Polak JF, Kronmal RA, Manolio TA, Burke GL, Wolfson SK, Jr. Carotid-artery intima and media thickness as a risk factor for myocardial infarction and stroke in older adults. Cardiovascular Health Study Collaborative Research Group. N Engl J Med. 1999;340:14-22.

---

## [Decision Letter · Decision Letter 1]

8 Nov 2022

Utility of carotid ultrasound on prediction of 1-year mortality in emergency department patients with neurological deficits: A 10-year population-based cohort study

PONE-D-22-19297R1

Dear Dr. Edward Pei-Chuan Huang,

We’re pleased to inform you that your manuscript has been judged scientifically suitable for publication and will be formally accepted for publication once it meets all outstanding technical requirements.

Kind regards,

Seung-Hwa Lee

Academic Editor

PLOS ONE

Additional Editor Comments (optional):

Reviewers' comments:

Reviewer's Responses to Questions

**Comments to the Author**

1. If the authors have adequately addressed your comments raised in a previous round of review and you feel that this manuscript is now acceptable for publication, you may indicate that here to bypass the “Comments to the Author” section, enter your conflict of interest statement in the “Confidential to Editor” section, and submit your "Accept" recommendation.

Reviewer #2: All comments have been addressed

2. Is the manuscript technically sound, and do the data support the conclusions?

Reviewer #2: Yes

3. Has the statistical analysis been performed appropriately and rigorously? 

Reviewer #2: Yes

4. Have the authors made all data underlying the findings in their manuscript fully available?

Reviewer #2: Yes

5. Is the manuscript presented in an intelligible fashion and written in standard English?

Reviewer #2: Yes

6. Review Comments to the Author

Reviewer #2: All the suggestions I made have been addressed. I do not have further comments for the authors.

7. PLOS authors have the option to publish the peer review history of their article (what does this mean?). If published, this will include your full peer review and any attached files.

Reviewer #2: **Yes: **Ivan Ferraz-Amaro

---

## [Editor Report · Acceptance letter]

21 Nov 2022

PONE-D-22-19297R1 

Utility of carotid ultrasound on prediction of 1-year mortality in emergency department patients with neurological deficits: A 10-year population-based cohort study 

Dear Dr. Huang:

I'm pleased to inform you that your manuscript has been deemed suitable for publication in PLOS ONE. Congratulations! Your manuscript is now with our production department. 

Kind regards, 

on behalf of

Dr. Seung-Hwa Lee 

Academic Editor

PLOS ONE